# When the Pandemic Will Be Over: Lots of Hope and Some Concerns

**DOI:** 10.3390/geriatrics7050116

**Published:** 2022-10-18

**Authors:** Domenico Azzolino, Giulia Carla Immacolata Spolidoro, Alessandra Mazzocchi, Carlo Agostoni, Matteo Cesari

**Affiliations:** 1Department of Clinical and Community Sciences, University of Milan, 20122 Milan, Italy; 2Geriatric Unit, Istituti Clinici Scientifici Maugeri IRCCS, 20138 Milan, Italy; 3Pediatric Intermediate Care Unit, Fondazione IRCCS Ca’ Granda Ospedale Maggiore Policlinico, 20122 Milan, Italy

**Keywords:** long COVID, COVID-19, nutrition, frailty, sarcopenia, public health, life course, intrinsic capacity, rehabilitation

## Abstract

The COVID-19 pandemic significantly threatened healthcare systems worldwide. The worst-hit population has been represented by older people with underlying chronic comorbidities, while children and adolescents developed mild or asymptomatic forms of the disease. However, certain medical conditions (i.e., obesity, respiratory, or neurological or immune disorders) may increase the risk for poor health outcomes even in young and middle-aged people. Beyond the direct negative effects of the infection, the pandemic posed several health challenges through an increase in psycho–social issues (i.e., anxiety, depression, fatigue, sleep alterations, loneliness, reduced assistance, and loss of income). Accordingly, the pandemic is negatively impacting the accumulation of the functional reserves of each individual, starting from early life. With the long-term effects of the pandemic to be seen in the coming years, clinicians must be prepared to manage such high clinical complexity of people they encounter, through the implementation of multidimensional and multidisciplinary interventions.

## 1. Introduction

The COVID-19 pandemic posed serious challenges to the healthcare systems worldwide and significantly changed our life. The worst-hit population during this pandemic is represented by older people with underlying chronic comorbidities. On the other hand, children and adolescents seem to develop asymptomatic or mild forms of the infection [1]. However, even younger people with certain underlying medical conditions (i.e., obesity, respiratory, or neurological or immune disorders) can experience adverse health outcomes including mortality [1,2]. From a life course perspective, early life is the sensitive period in which each person builds up their biological reserves. Accordingly, adverse early life exposures can influence human health in the long term since the rate of functional decline observed during aging depends, beyond later life conditions, on the peak of reserves attained during early development [3]. In other words, socio-historical negative events, such as the COVID-19 pandemic, can have a negative impact on health trajectories through various environmental factors determining a decreased accumulation of biological reserves. According to life course theory, there are developmental turning points in which the accumulation of disadvantages can deflect resilience trajectories [4]. The same risk of poor outcomes from COVID-19 infection depends on the accumulation of prior exposure to health-related stressors determining later-life conditions (i.e., multimorbidity, frailty, and psycho–social problems). The management of COVID-19 worldwide has thus far concentrated more on treating the acute phase of the illness, but, compared to the beginning of the pandemic, it is now becoming increasingly evident that, regardless of age, COVID-19-infected subjects may experience debilitating long-COVID symptoms for many months after developing the disease [5]. Unfortunately, even the most advanced healthcare systems, which were originally based and remain centered on the “single disease” approach, have been found largely unprepared to manage the high clinical complexity of frail older people, as well as the needs of the young categories faced with COVID-19. Despite older adults representing the largest portion of the population affected by COVID-19, they received relatively little attention. Indeed, some ageism phenomena have been seen in our societies, with the impression that COVID-19 is only a problem of old age [6]. As a result, chronological age has been frequently used as the unique criterion for allocating resources to support the care of patients with COVID-19. Serious consequences from a COVID-19 infection can be observed in children too, such as the multisystem inflammatory syndrome in children (MIS-C) [5]. 

The secondary effects of the COVID-19 pandemic have also been large. People of all age groups have suffered from the situation both socially and psychologically [7]. In particular, containment measures to limit the spread of the virus posed an important threat to older people’s health (i.e., reduced mobility and assistance, worsening of depressive symptoms, sleep disorders, nutritional issues). Some services have changed or been interrupted, including outpatient clinics and preventive programs. Despite video conferencing aids and telerehabilitation strategies having been implemented, it should be noted that older people are frequently not very confident with new technologies and may present visual and hearing problems challenging the interaction via these channels. It has been widely reported that older adults need assistance to learn a new technology. Furthermore, factors such as advanced age, lower education and income, and poor health status have been described as barriers in learning a new technology during the COVID-19 pandemic [8]. Indeed, addressing some of these barriers (i.e., sociodemographic and clinical factors) can be helpful in the adoption of new technologies among older adults. The COVID-19 pandemic also introduced some challenges to clinical research projects, which have been often re-shaped or even interrupted [9]. 

Finally, in the context of COVID-19, other elements are gaining increasing importance in the management of the pandemic. To name a few, nutritional status has been indicated as an important mediator of the infection, modulating the susceptibility and severity of the disease [3,4,5]. At the same time, vaccination campaigns are widely demonstrating their efficacy in preventing severe disease and death. However, the problem of vaccine inequity between developed and developing countries is yet to be resolved, holding up the chances to reach global immunization [10,11]. In the perspective of moving closer to the end of the pandemic, topics such as nutrition and global immunization as well as rehabilitative strategies should therefore be discussed and implemented nationally and internationally.

## 2. Psycho–Social Impact

The psycho–social impact of the COVID-19 pandemic should not be overlooked. The implementation of prolonged containment measures adopted to limit the spread of the virus negatively impacted the lives of everyone resulting in anxiety, fear of being infected, loneliness, increased depressive symptoms, sleep alterations, reduced assistance, loss of income, and fatigue [12,13]. Furthermore, in this case, older people represented the worst-hit population since during the various lockdowns they experienced a reduction in assistive procedures, limited access to grocery shopping as well as reduced support for shopping and cooking [14], just to name a few. To date, sleep disorders and depressive symptoms are already too-prevalent conditions in older people and have been exacerbated by such drastic countermeasures. The COVID-19 pandemic also increased the prevalence and worsening of some other conditions such as agitation, anxiety, and delirium frequently observed in older people, especially in those with cognitive deterioration [15,16]. The message that the most vulnerable group to adverse outcomes from COVID-19 infection is represented by older people with underlying comorbidities created a noticeable sense of fear among older adults. Nursing home residents also experienced dramatic behavioral changes, since for several months the structures in which they lived were completely closed to visitors in order to protect them from the infection. As already anticipated, younger people are no strangers to the psychological and social burden that has resulted from COVID-19 pandemic. Recent studies investigating the long-term effect of COVID-19 on children and adolescents have surprisingly suggested that, for all outcomes related to the psychological and social sphere, young subjects who did not develop COVID-19 reported even more frequent problems than their counterparts who instead developed the disease [17,18]. However, the two extremes of life present substantial differences in terms of the needed psycho–social support. To date, older people, despite needing some forms of support, can frequently experience loneliness and social isolation. Cognitive as well as financial problems are also more prevalent in older adults than in their younger counterparts.

During the COVID-19 pandemic, the rates of domestic violence (i.e., sexual, psychological, and physical abuse) markedly increased, with children, women, and older people representing the most vulnerable groups [19,20]. From a life course perspective, psychological and social exposures to adverse events, starting from early life, have long-term sequelae including declined mental and physical capacities [21]. In particular, adult cognition is strongly determined by educational attainment as well as childhood cognitive ability [22] and most of mental health problems are determined during adolescence and early adulthood [23]. Indeed, it should not be overlooked that the disruption of educational programs also poses a threat to educational attainment, which is widely recognized as a risk factor for cognitive deterioration later in life [24]. Furthermore, socioeconomic disadvantage across the life course has been associated with accelerated aging, decline in muscle strength, and the occurrence of multiple morbidities as well as reduced access to healthcare [25]. 

### Fatigue and COVID-19

The World Health Organization (WHO) recently released a document introducing the term “pandemic fatigue” to identify the symptom resulting from an unprecedented and prolonged public health crisis. In this document, the WHO defined pandemic fatigue as “a demotivation to follow recommended protective behaviors, emerging gradually over time and affected by a number of emotions, experiences and perceptions” [26]. Fatigue is one of the most complained-about symptoms by older people and has been also reported as the most prevalent and long-lasting symptom after SARS-CoV-2 infection [27,28]. There are several mechanisms that can be involved in the manifestation of fatigue, such as inflammation and mitochondrial dysfunction, autonomic nervous system abnormalities, poor nutritional status, and sleep alterations. Interestingly, all these mechanisms are also observed in the context of COVID-19 [29]. Fatigue may therefore be envisioned as a clinical indicator of an underlying biological abnormality [30,31]. However, the subjectivity of the symptom and the lack of a gold standard measure for fatigue assessment contribute to its poor consideration in the clinical practice [29]. It should be noted that fatigue can be perceived even in middle-aged and young people following stressful conditions such as containment measures. In fact, everyone suffered from these drastic measures, changing their behaviors such as outdoor activities or spending time with their peers and other family members towards unhealthy behaviors (i.e., increased sitting times, watching TV, video gaming and computer use, junk food consumption). Depression, sleep alterations, and fatigue frequently coexist and interact, creating a vicious circle and thus exposing the older person to adverse outcomes [32]. Indeed, it is pivotal to also properly consider other components related to the psycho–social sphere which may affect individual health status.

## 3. Impact on Nutritional Status

Nutritional status has been indicated as a pivotal factor mediating COVID-19 susceptibility and severity [14,33,34,35]. Older people with underlying chronic comorbidities are at high risk for malnutrition and, consequently, adverse clinical outcomes in COVID-19 infection [36]. The relationship between nutritional status and SARS-CoV-2 infection looks to be bidirectional (Figure 1). On the one hand, poor nutritional status can determine an increased susceptibility and severity of the infection even in young persons, despite the most vulnerable group remaining composed of older people [1]. In fact, both malnutrition (i.e., undernutrition) and obesity have been associated with the worst outcomes in COVID-19 patients [33,35,36,37]. In particular, in older people undernutrition could determine an augmented susceptibility and severity of the infection since it is associated with defective immune response, increased risk of infections, loss of muscle mass and strength, delayed wound healing, and prolonged hospital stay [38,39,40]. On the opposite side, an obesity background could predispose to the most severe consequences of the disease through several mechanisms including inflammation and decreased immune response to viral infections. Obesity status is usually associated with type-2 diabetes, hypertension, and cardiovascular disease, all of which have been indicated as predictive of poor outcomes in COVID-19 infection [41]. Additionally, a high expression of angiotensin-converting enzyme 2 (ACE2) receptor, which is responsible for the SARS-CoV-2 virus entry in the organism, has been largely found in the adipose tissue. This finding could probably explain the increased susceptibility to the infection of those people exhibiting greater adiposity [42,43]. Obesity is often overlooked in older people, and can present an excess of body fat, concomitant to low muscle mass, even in the context of a normal/overweight body mass index (i.e., sarcopenic obesity) [44]. On the other hand, COVID-19 is associated with loss of taste and smell and gastrointestinal alterations (i.e., nausea vomiting, diarrhea), characteristic of the so-called “anorexia of aging”, contributing to a reduction in dietary intake [45]. Furthermore, in older people, COVID-19 infection could predispose to myalgias and muscle wasting through an increased inflammatory state and thus higher catabolism [45]. On the other hand, COVID-19 through the alteration of nutritional status during early life may negatively impact growth and development, the period in which each person builds up its functional capacities. Therefore, following a life course approach, during this period the attainment of the individual functional and structural reserve (especially in terms of muscle mass and strength) should be maximized in order to minimize the rate of functional decline during older life [46]. 

Lockdown measures in many countries resulted in a change in healthy eating and exercising. There has been an increase in the consumption of ultra-processed convenience foods (characterized by high content of saturated fatty acids, sugar, and salt), meal frequency, snacking, and alcohol intake parallel to the decrease in the consumption of fresh and natural healthy food (i.e., fruit and vegetables) rich in vitamins and minerals, and increase in sedentary behaviors (i.e., increased sitting times) [14,47]. These changes in health behaviors have been exacerbated by the food price increase consequent to the disruption of agricultural production, seasonal labor, and fuel price increases. The recent war in Ukraine further augmented barriers to proper nutrition, including outside the war zone, creating a global food crisis increasing the risks of famine and food insecurity [48]. In fact, many population groups at risk of malnutrition have been severely affected by the joint effects of conflicts, climate change, and the COVID-19 pandemic [48]. Food insecurity has been associated with multimorbidity (i.e., the coexistence of two or more chronic diseases) in older people, especially in low- and middle-income countries (LMICs) [49]. In turn, the presence of multiple chronic conditions may lead to food insecurity, probably because of the financial strain (i.e., expenditures for medication, rehabilitation, transportation), creating a vicious circle [50,51]. The interruption of school meal programs further hampered proper and healthy nutrition during early life, especially in those families with poor socioeconomic status [52].

The problem of undernutrition (i.e., stunting, wasting, and underweight) has been thus exacerbated by COVID-19 pandemic starting from early life [53], but which may have intergenerational and long-term health consequences (i.e., altered body composition, poor cognitive function, poor organ function, adult chronic diseases), especially in LMICs [54]. On the other hand, in Western developed countries, the COVID-19 pandemic is steadily increasing the prevalence of overweight and obesity, which have been largely associated with long-term risk of metabolic diseases (i.e., diabetes, hypertension, cardiovascular diseases, cancer) [53,55,56,57].

Several micronutrients have been associated with poor outcomes during viral infections [58]. All the vitamins and minerals needed by people can be ensured by eating a healthy and balanced diet. However, supplementation should be considered in those people who are malnourished or at risk of deficiency [36]. Vitamin D, for example, is pivotal for bone health, and low levels of vitamin D have been associated with multiple chronic diseases [59] as well as reduced muscle mass [60]. Vitamin D (i.e., cholecalciferol) is mainly synthesized in the skin after sun exposure. Indeed, during the various lockdowns, people experienced less sun exposure, with a negative impact on vitamin D status [47,61].

Indeed, nutritional status needs to be strictly and routinely monitored at every change in health status.

## 4. Vaccination and Immunity

Since the end of December 2020 when it started, COVID-19 vaccination is ongoing and, in many countries, also involved children and adolescents. Vaccination priority has been given to older adults and frail, polymorbid individuals, who represent the category at risk of the most severe consequences of the disease. With aging, there is a progressive decline in immune function (i.e., immunosenescence) which can be associated with a reduced ability to respond to infections [62] and can limit the effectiveness of vaccinations [63]. However, COVID-19 vaccines have been indicated as effective in reducing hospitalizations and mortality in older people despite the need for booster doses to optimize response [62,64,65]. Accordingly, older people are among those who profited most from vaccination campaign in terms of adverse outcomes. Children seem to develop less severe forms of the disease. However, they can become infected and can potentially spread the virus through the community, in particular to older persons [66]. Indeed, to limit community transmission of the virus and to protect frail individuals, a massive vaccination campaign is ongoing, also involving children and adolescents. Unfortunately, some individuals remain unvaccinated and are those experiencing the worst outcomes including hospitalization, intensive care, mechanical ventilation, and death [67]. Additionally, SARS-CoV-2 variants characterized by increased transmission capacity have emerged and raised concerns in the scientific community, especially for a possible lower efficacy of vaccines against these new forms of the virus. Recently, there has been a net decrease in the mean age of infected people that can spread the virus to more vulnerable groups (i.e., older people with comorbidities) [68]. It has also been outlined that high vaccination coverage may not be sufficient to dampen the spread of the virus, since low coverage in local groups can result in outbreaks [69,70]. Indeed, it is important to carry out vaccination campaigns with additional or booster doses, starting from frail individuals but extending it to all population groups including children and adolescents. On this note, supporting vaccination campaigns even in developing countries would be equally important.

## 5. Demographic Projections and Frailty

Given the high mortality rates and the health-related negative effects of the pandemic, it should be not overlooked that COVID-19 effects on demographic projections are yet to be determined. A decline in the older population created by the excess deaths during the COVID-19 pandemic has been seen [71]. Furthermore, the health disparity resulting from a high prevalence of COVID-19 cases and death among ethnic minorities (e.g., Afro-Americans), may be partially explained by the high prevalence of obesity and diabetes as well as limited access to healthy food choices and to healthcare in these populations [72,73]. As mentioned above, the long-term consequences of the COVID-19 pandemic may negatively impact the accumulation of the biological reserves of the individual across the lifespan starting from early life. Indeed, in the coming years, we could see a high prevalence of pandemic-related negative conditions increasing the risk of frailty. Interventions against frailty have long been recognized as an opportunity to re-organize the traditional models of care in order to manage the multiple problems of older people [74]. In other words, there is a need for multidimensional and interdisciplinary interventions targeted at the functional capacities of the individual. In this context, the multidimensional approach of pediatric and geriatric specialties, encompassing the two extremes of life, can be envisioned as a model to properly address the multi-system (long-lasting) problems seen within the pandemic in a life course perspective. Integrating geroscience with life course epidemiology can be envisioned as a promising perspective to capture dynamic modifications of health trajectories. 

## 6. Lifestyle Interventions 

The COVID-19 pandemic is continuously posing a serious threat to the health of everyone both directly and indirectly. The sequelae of COVID-19 depend on the extension and severity of the infection in the various organ systems [75]. Indeed, the term “Long COVID” has been coined to define the persistence of symptoms and other medical complications involving multiple organs and systems (i.e., respiratory, cardiovascular, neurological, gastrointestinal, and musculoskeletal systems), which can last for several weeks or months [76,77]. Indeed, there is a need for prompt and multidisciplinary interventions to properly manage both the acute and the long-lasting effects of this pandemic. In both cases, environmental changes play a pivotal role. According to experts’ recommendations, nutritional assessment and management should be envisioned as an integral part of the continuum of care for COVID-19 patients [36,75]. In other words, nutritional status needs to be assessed in all infected patients, since poor nutrition has been associated with the worst prognosis of the disease. Particular attention should be paid to older adults with underlying comorbidities, who are those at higher risk for malnutrition and adverse clinical outcomes [36]. It is noteworthy that nutritional needs are largely different at the two extremes of life (i.e., early life and older life). 

Following a life course approach, it is important to optimize nutrition starting from early life. However, no specific guideline on nutritional recommendations for infected children and adolescents has been released. During early life, dietary needs to prevent malnutrition are quite variable according to age, weight, and disease severity. Some documents stressed the importance of breastfeeding during early life with a special focus on COVID-19 infection since breastfeeding can protect against infections and respiratory diseases. In fact, breast milk is a source of a range of antibodies, enzymes, and hormones with a clear health benefit [78,79]. In particular, the World Health Organization stated that women with COVID-19 can breastfeed [80,81]. The Centers for Disease Control and Prevention (CDC) highlighted that despite COVID-19 not being detected in breast milk, it is unknown if infected mothers can spread COVID-19 virus during breastfeeding. Indeed, the CDC advised wearing a face mask and to wash hands before breastfeeding [82]. Other recommendations include a reduction in the intake of salt, sugar, and in general ultra-processed food rich in saturated fatty acids, as well as the strict monitoring of micronutrient status (i.e., zinc and vitamins C, A, and D), especially for their contribution to immune function [78]. For what concerns adults and especially older adults, it is important to ensure adequate intake of energy and proteins because of increased catabolism. In particular, the European Society for Clinical Nutrition and Metabolism (ESPEN) claimed that energy needs could be evaluated through indirect calorimetry, if safely available (i.e., sterility of the measurement system) [36]. However, indirect calorimetry is not available in most settings. Indeed, ESPEN recommends assessing energy needs using alternative measures (i.e., prediction equations or weight-based formulae) when it is not possible to use indirect calorimetry. In particular, it is recommended to have an energy provision of 27–30 kcal/kg of body weight/day for polymorbid patients aged >65 years according to nutritional status physical activity level, disease, and preferences [36]. However, given the risk of refeeding syndrome, the target of 30 kcal/kg of body weight/day should be slowly attained in severely undernourished patients. In older adults, a protein intake of at least 1.0 g/kg of body weight/day should be ensured, augmenting it to 1.2–1.5 g/kg of body weight/day in the presence of acute or chronic diseases and reaching up to 2.0 g/kg of BW/day in the presence of highly catabolic conditions. However, given that older people may present several issues (i.e., early satiety, decreased appetite, delayed gastric emptying) that can hamper the achievement of these nutritional goals, supplementation strategies should be considered. 

Beyond nutrition, it is pivotal to pay particular attention to physical activity. Prolonged bed rest and immobilization due to hospitalization for COVID-19 infection is associated with multi-system impairment (i.e., cognition, locomotion, vitality) and therefore functional decline and adverse outcomes [83]. This functional decline, even if to a lesser extent, is seen also in those people confined at home who may inevitably experience muscle and bone decline, cognitive deterioration, decreased ability to resist viral infection as a consequence of physical inactivity, unhealthy eating, and psycho–social factors (i.e., isolation, loneliness), augmenting the risk for frailty [84]. 

Decreased physical activity during the various lockdowns has also been associated with increased anxiety, mental health issues, and anergia (i.e., low energy levels) [85,86]. Indeed, it is pivotal to promote physical activity even with activities performed at home (i.e., home workouts, online exercise programs) in the case of lockdowns and confinement [85]. It is also important to reduce sitting times during confinement and smart working to reduce a sedentary lifestyle. Physical activity, even if performed indoors, may be also helpful in counteracting the psychological impact of this pandemic [15]. For what concerns hospitalized patients, early mobilization strategies based on different movements (i.e., from passive to resistive movements up to activities) should be implemented immediately after clinical stabilization [87]. A supportive environment, both physical and social, is essential to address older people’s unmet needs created by the current pandemic. Strategies addressing food insecurity, such as food banks, should be largely implemented. 

## 7. Conclusions and Future Perspectives

Clinicians must be prepared to deal with the high clinical complexity of people they will encounter. Indeed, there is a need for a multidisciplinary and multidimensional approach to manage the complexity of the individuals burdened by the pandemic. In this context, holistic processes, such as the comprehensive geriatric assessment (CGA) [88] used in the geriatric setting to personalize interventions according to medical, psychological, and functional problems, are also needed in other age groups. It is also important to dynamically identify health tipping points across life and not at one point in time, in order to prevent, at an early stage, long-term negative consequences. Chronological age must no longer be used as a criterion to provide care for older people beyond this pandemic. Proper lifestyle interventions, encompassing nutrition and physical activity and taking into account social and psychological aspects, must be urgently prioritized. 

## Figures and Tables

**Figure 1 geriatrics-07-00116-f001:**
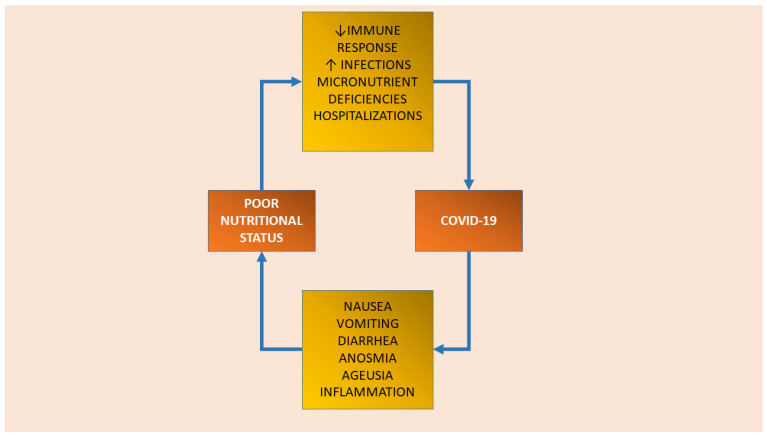
The bidirectional relationship between poor nutritional status and COVID-19 infection.

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
