# Peer review of "When the Pandemic Will Be Over: Lots of Hope and Some Concerns"

_geriatrics, 2022, doi:10.3390/geriatrics7050116_

Round 1

Reviewer 1 Report

The authors have written a timely paper identifying health challenges during the evolving COVID-19 pandemic situation. In addition, the authors pointed out several challenges to be noted, such as Psycho-social impact, Nutritional status, Vaccination, Demographic projections and frailty. In my opinion, this paper can be a contribution to the current literature by addressing several limitations. The main comment for the authors to consider is to not only propose the challenges, but also provide more perspective on how to address the barriers to these challenges.

  1. Digging more into the existing barriers to the aforementioned challenges.

For example, the authors stated, “Despite video conferencing aids and telerehabilitation strategies having been implemented, it should be noted that older people are frequently not very confident with new technologies and may present visual and hearing problems challenging the interaction via these channels.”

According to this, the authors can cite papers such as:

Li W, Ornstein KA, Li Y, Liu B. Barriers to learning a new technology to go online among older adults during the COVID-19 pandemic. J Am Geriatr Soc. 2021 Nov;69(11):3051-3057. doi: 10.1111/jgs.17433. Epub 2021 Aug 29. PMID: 34409589; PMCID: PMC8446986.

Then the authors can expand the discussion on how addressing potential barriers can alleviate the challenges.

  1. Throughout the paper, the authors emphasize that geriatrics and pediatrics both need to be prepared for those health challenges. For example, in the section on Psycho-social impact, the authors first mentioned, “Also in this case, older people represented the worst-hit population since during the various lockdowns they experienced a reduction of assistive procedures...” Later at the end of this section, the authors also mentioned, “As already anticipated, children and adolescents are no strangers to the psychological and social burden that has resulted from COVID-19 pandemic.” It is true that both children/adolescents and older adults may be suffering from the psycho-social impact. However, the authors may consider elaborating on how the dramatic differences between children/adolescents and older adults relative to the management guidelines on the needed psychosocial support.

  1. Similarly, in the section on nutritional status, the authors could expand some discussion on the differences between children/adolescents and older adults, as their needs are largely different.

  1. In the section on demographic projections and frailty, when talking about excess death due to the pandemic, the authors may consider adding discussion on the disproportionate impact of the pandemic on different races. Given that racial minorities experienced a larger burden of COVID-19 infections and deaths, it may be interesting for the reader to learn more about the disparities. For example, consider to reference:

Vasquez Reyes M. The Disproportional Impact of COVID-19 on African Americans. Health Hum Rights. 2020;22(2):299-307.

Author Response

Reviewer 1

The authors have written a timely paper identifying health challenges during the evolving COVID-19 pandemic situation. In addition, the authors pointed out several challenges to be noted, such as Psycho-social impact, Nutritional status, Vaccination, Demographic projections and frailty. In my opinion, this paper can be a contribution to the current literature by addressing several limitations. The main comment for the authors to consider is to not only propose the challenges, but also provide more perspective on how to address the barriers to these challenges.

Digging more into the existing barriers to the aforementioned challenges.

For example, the authors stated, “Despite video conferencing aids and telerehabilitation strategies having been implemented, it should be noted that older people are frequently not very confident with new technologies and may present visual and hearing problems challenging the interaction via these channels.”

According to this, the authors can cite papers such as:

Li W, Ornstein KA, Li Y, Liu B. Barriers to learning a new technology to go online among older adults during the COVID-19 pandemic. J Am Geriatr Soc. 2021 Nov;69(11):3051-3057. doi: 10.1111/jgs.17433. Epub 2021 Aug 29. PMID: 34409589; PMCID: PMC8446986.

Then the authors can expand the discussion on how addressing potential barriers can alleviate the challenges. (supportive environment)

Response: Thank you for the time spent reviewing our manuscript and for your valuable comments and suggestions. We have revised the manuscript accordingly. We have tried to focus more on how to address the barriers created by the pandemic, also implementing your great suggestions.

Throughout the paper, the authors emphasize that geriatrics and pediatrics both need to be prepared for those health challenges. For example, in the section on Psycho-social impact, the authors first mentioned, “Also in this case, older people represented the worst-hit population since during the various lockdowns they experienced a reduction of assistive procedures...” Later at the end of this section, the authors also mentioned, “As already anticipated, children and adolescents are no strangers to the psychological and social burden that has resulted from COVID-19 pandemic.” It is true that both children/adolescents and older adults may be suffering from the psycho-social impact. However, the authors may consider elaborating on how the dramatic differences between children/adolescents and older adults relative to the management guidelines on the needed psychosocial support.

Similarly, in the section on nutritional status, the authors could expand some discussion on the differences between children/adolescents and older adults, as their needs are largely different.

Response: Thank you for your comment. We modified this part trying to highlight the differences between the two extremes of life even from a life course perspective.

In the section on demographic projections and frailty, when talking about excess death due to the pandemic, the authors may consider adding discussion on the disproportionate impact of the pandemic on different races. Given that racial minorities experienced a larger burden of COVID-19 infections and deaths, it may be interesting for the reader to learn more about the disparities. For example, consider to reference:

Vasquez Reyes M. The Disproportional Impact of COVID-19 on African Americans. Health Hum Rights. 2020;22(2):299-307.

Response: Thank you for your suggestion. We modified this part accordingly.

Reviewer 2 Report

The work provides a perspective on ageism in the context of the pandemic and other shortcoming of the strategies to confront it such as the conception of ‘a single disease’ focused on the virus vs current vision of a more complex and multidimensional scenario that is worsened by the long-COVID symptoms The subsections provide the author’s perspective on the psychosocial impact, nutritional status, vaccination, demographic projections and frailty, life intervention, and final conclusions that highlight the need to be prepared for such a complexity

The titles of the subsections are very generic (such as psycho-social impact) and cover a bast concept in few parapraphs. From all the items mentioned, some are of interest. For instance, in the 'pandemic fatigue' is an important issue. I'd recommend to provide a subtitle (for the subsection) that summarizes, as a kind of a home-take message, the key aspects refering to older people.

Section 3, Nutritional status is a brief but good summary of the main aspects. However, the part refering to the older people is missing, despite this perspective has been submitted to a Geriatrics journal . The same accounts for the subsequent sections,where only minor aspects are indicated. In a general manner, the contents refer to the scenario /topic, but are written in a manner that is not focused on the target population (as the journal is Geriatrics, one would expect that). Section 4, has only 1 sentence (lines 162-164) for the impact on older people. Section 5 is correct, as it refers to demographics and it is more easy to find the focus on the frail population.
Section 6 contains sentences such as 202-204 and subsequent that refer to the lack of recommendations for pediatric populations (children/adolescents) that suggest that the Ms has been written in a generic manner (for all the population) but not focused on older people to get a better understanding. Section 7 reinforces this perception when starts with a 'Geriatrics and pediatrics but also...'
In my opinion, for the way the perspective is written is not in the scope of the Geriatrics journal, but a more 'general population', so the authors should reconsider all the contents, enrich them with geriatric scopes. I consider the other population groups be relevant to mention only if there is a comparative analysis of discussion to illustrate how 'age'-personalized impact or strategies should be implemented, as the different groups were impacted or behavied in a different way.

Title
‘When the pandemic will be over” Is it a question? It will suit to be so.

Line 47, The indirect effects of the COVID-19 pandemic have also been large. Children and

Please, refer to secondary impact of COVID-19.

I strongly suggest to do a graphical abstract

Author Response

Reviewer 2

The work provides a perspective on ageism in the context of the pandemic and other shortcoming of the strategies to confront it such as the conception of ‘a single disease’ focused on the virus vs current vision of a more complex and multidimensional scenario that is worsened by the long-COVID symptoms The subsections provide the author’s perspective on the psychosocial impact, nutritional status, vaccination, demographic projections and frailty, life intervention, and final conclusions that highlight the need to be prepared for such a complexity

The titles of the subsections are very generic (such as psycho-social impact) and cover a bast concept in few parapraphs. From all the items mentioned, some are of interest. For instance, in the 'pandemic fatigue' is an important issue. I'd recommend to provide a subtitle (for the subsection) that summarizes, as a kind of a home-take message, the key aspects refering to older people.

Response: Thank you very much for reviewing our manuscript and for your helpful comments. We have carefully considered your suggestions. We created a specific subsection for fatigue (i.e., 2.1. Fatigue and COVID-19). We hope we have been able to address what you meant. 

Section 3, Nutritional status is a brief but good summary of the main aspects. However, the part refering to the older people is missing, despite this perspective has been submitted to a Geriatrics journal. The same accounts for the subsequent sections, where only minor aspects are indicated. In a general manner, the contents refer to the scenario /topic, but are written in a manner that is not focused on the target population (as the journal is Geriatrics, one would expect that). Section 4, has only 1 sentence (lines 162-164) for the impact on older people. Section 5 is correct, as it refers to demographics and it is more easy to find the focus on the frail population.

Section 6 contains sentences such as 202-204 and subsequent that refer to the lack of recommendations for pediatric populations (children/adolescents) that suggest that the Ms has been written in a generic manner (for all the population) but not focused on older people to get a better understanding. Section 7 reinforces this perception when starts with a 'Geriatrics and pediatrics but also...'

In my opinion, for the way the perspective is written is not in the scope of the Geriatrics journal, but a more 'general population', so the authors should reconsider all the contents, enrich them with geriatric scopes. I consider the other population groups be relevant to mention only if there is a comparative analysis of discussion to illustrate how 'age'-personalized impact or strategies should be implemented, as the different groups were impacted or behavied in a different way.

Response: Thank you for your comments. We revised the manuscript accordingly by focusing more on the geriatric population and also emphasizing a life course approach. We hope that our revision is now acceptable. 

Title

‘When the pandemic will be over” Is it a question? It will suit to be so.

Response: Thank you for your suggestion. We changed the title accordingly.

Line 47, The indirect effects of the COVID-19 pandemic have also been large. Children and

Please, refer to secondary impact of COVID-19.

Response: Thank you. We revised this part according to your suggestion.

I strongly suggest to do a graphical abstract

Response: Thank you. We added a graphical abstract.

Reviewer 3 Report

The authors submitted an extensive perspective manuscript entitled “When the pandemic will be over: Lots of hope and some concerns”. Unfortunately, in my opinion the manuscript is difficult to follow and shows some concepts in a repetitive manner. Overall, I believe that the manuscript intends to approach to too many things and fails to show clear key messages.    

Since the manuscript has been submitted to geriatrics, I do not understand why the manuscript also focuses in the pediatric population.

I found a marked overlapping among the contents referred to nutritional status and lifestyle interventions

Vaccinations section seems out of place in this manuscript in relation to the rest of the contents

Author Response

Reviewer 3

The authors submitted an extensive perspective manuscript entitled “When the pandemic will be over: Lots of hope and some concerns”. Unfortunately, in my opinion the manuscript is difficult to follow and shows some concepts in a repetitive manner. Overall, I believe that the manuscript intends to approach to too many things and fails to show clear key messages.   

Since the manuscript has been submitted to geriatrics, I do not understand why the manuscript also focuses in the pediatric population.

Response: Thank you for revising our manuscript. We revised the manuscript by focusing more on the geriatric population and emphasizing a life course approach. We refer to early life in the manuscript since, in a life course perspective, environmental factors occurring during critical or sensitive periods of life can have long-term negative effects by deflecting health trajectories. 

I found a marked overlapping among the contents referred to nutritional status and lifestyle interventions

Response: Thank you for your comment. We revised this part. We separated the two paragraphs in order to differentiate the impact on nutritional status from the interventions.

Vaccinations section seems out of place in this manuscript in relation to the rest of the contents

Response: Thank you for your comment. We modified this paragraph also by contextualizing it to immune function in older adults.

Round 2

Reviewer 1 Report

I thank the authors for their response to my comments. I believe the paper has been much improved and is suitable for publication.

Reviewer 3 Report

The manuscript has improved in its present version

I have no major objections